# Nematicidal Coumarins from *Cnidium monnieri* Fruits and *Angelica dahurica* Roots and Their Physiological Effect on Pine Wood Nematode (*Bursaphelenchus xylophilus*)

**DOI:** 10.3390/molecules28104109

**Published:** 2023-05-15

**Authors:** Jiale Feng, Chenglei Qin, Xiaohong Liu, Ronggui Li, Chao Wang, Chunhan Li, Guicai Du, Qunqun Guo

**Affiliations:** 1College of Life Sciences, Qingdao University, Qingdao 266071, China; 18053687042@163.com (J.F.); 18863635102@163.com (C.Q.); lrg@qdu.edu.cn (R.L.); wangchao6903199@163.com (C.W.); chunhanlee1226@163.com (C.L.); 2School of Pharmacy, Qingdao University, Qingdao 266071, China; liuxiaohongqd@126.com

**Keywords:** *Bursaphelenchus xylophilus*, nematicidal activity, *Cnidium monnieri* fruits, *Angelica dahurica* roots, coumarins, cindimine

## Abstract

Pine wood nematode (PWN), *Bursaphelenchus xylophilus*, is a major pathogen of pine wilt disease (PWD), which is a devastating disease affecting pine trees. Eco-friendly plant-derived nematicides against PWN have been considered as promising alternatives to control PWD. In this study, the ethyl acetate extracts of *Cnidium monnieri* fruits and *Angelica dahurica* roots were confirmed to have significant nematicidal activity against PWN. Through bioassay-guided fractionations, eight nematicidal coumarins against PWN were separately isolated from the ethyl acetate extracts of *C. monnieri* fruits and *A. dahurica* roots, and they were identified to be osthol (Compound **1**), xanthotoxin (Compound **2**), cindimine (Compound **3**), isopimpinellin (Compound **4**), marmesin (Compound **5**), isoimperatorin (Compound **6**), imperatorin (Compound **7**), and bergapten (Compound **8**) by mass and nuclear magnetic resonance (NMR) spectral data analysis. Coumarins **1**–**8** were all determined to have inhibitory effects on the egg hatching, feeding ability, and reproduction of PWN. Moreover, all eight nematicidal coumarins could inhibit the acetylcholinesterase (AChE) and Ca^2+^ ATPase of PWN. Cindimine **3** from *C. monnieri* fruits showed the strongest nematicidal activity against PWN, with an LC_50_ value of 64 μM at 72 h, and the highest inhibitory effect on PWN vitality. In addition, bioassays on PWN pathogenicity demonstrated that the eight nematicidal coumarins could effectively relieve the wilt symptoms of black pine seedlings infected by PWN. The research identified several potent botanical nematicidal coumarins for use against PWN, which could contribute to the development of greener nematicides for PWD control.

## 1. Introduction

Pine wilt disease (PWD), as a devastating conifer disease, has caused severe worldwide damage to susceptible pine species and even native forest ecosystems, especially in East Asia and Europe [1,2,3]. The pine wood nematode (PWN), *Bursaphelenchus xylophilus*, is the causal agent of PWD, and it has been listed as one of the top 10 plant-parasitic nematodes based on its economic and scientific importance [4,5]. So far, PWD control has become an urgent problem. Among various control tactics for PWD prevention, trunk injection of powerful nematicides against PWN remains one of the most effective, sustainable, and reliable strategies [3,6,7]. However, commonly used synthetic nematicides have garnered more and more concerns, regarding drug resistance, resulting from repeated applications, toxicity to non-target organisms, and loss of biodiversity [1,8,9,10,11]. Hence, there is an ever-increasing need to search for safer and eco-friendly nematicides to prevent PWN infection [8,9]. A broad spectrum of bioactive plant secondary metabolites, generated by plants to resist attackers over their lifetime, have been screened for their use as pesticides or model compounds for chemically synthesized derivatives with enhanced insecticidal activity and eco-friendliness [12,13], and a significant number of studies have focused on phytochemical-based strategies for nematode control [14,15,16]. To date, a growing number of nematicidal phytochemicals used against PWN have been evaluated as potential nematicides for PWD management and include plant essential oils, alkaloids, coumarins, flavonoids, and glycosides [3,17]. However, few nematicidal phytochemicals are developed into commercial nematicides due to a lack of research on their modes of action and mechanisms, and further work on plant-based nematicides is needed.

In this study, we screened two nematicidal botanical extracts, from *C. monnieri* fruits and *A. dahurica* roots, against PWN; both extracts are used as Chinese traditional herbal medicine and are rich in coumarins [18,19,20,21,22,23]. Several nematicidal coumarins were identified, and their effects on the egg hatching, feeding ability, reproduction, pathogenicity, and key enzymes of PWN were investigated, which facilitates the development of new plant-derived nematicides.

## 2. Results

### 2.1. Nematicidal Activity of Extracts

The ethanol extracts of *C. monnieri* fruits and *A. dahurica* roots both showed significant nematicidal activity against PWN, causing corrected mortalities of 77.93% and 84.64%, respectively, at 1 mg/mL in 72 h (Table 1 and Table 2). Furthermore, the ethyl acetate extracts of *C. monnieri* fruits and *A. dahurica* roots were much more active, with corrected mortalities of 96% and 86.15% at 1 mg/mL in 72 h, than their aqueous extracts (Table 1 and Table 2).

Among the 18 fractions obtained by column chromatography from the ethyl acetate extract of *C. monnieri* fruits, Fr. 1, Fr. 2, and Fr. 3 showed high nematicidal activity against PWN, with corrected mortalities of 93.92%, 91.22%, and 69.6%, respectively, at 500 μg/mL in 72 h (Table 3). In the 16 chromatographed fractions from the ethyl acetate extract of *A. dahurica* roots, Fr. 5, Fr. 6, and Fr. 7 demonstrated strong nematicidal activity, with corrected mortalities of 70.87%, 74.74% and 79.65%, respectively, at 500 μg/mL in 72 h (Table 4).

### 2.2. Identification of Nematicidal Compounds

Through nematicidal activity-guided silica gel chromatography, eight active compounds were isolated from the extracts of *C. monnieri* fruits and *A. dahurica* roots, and they were identified as osthol (Compound **1**), xanthotoxin (Compound **2**), cindimine (Compound **3**), isopimpinellin (Compound **4**), marmesin (Compound **5**), isoimperatorin (Compound **6**), imperatorin (Compound **7**), and bergapten (Compound **8**) (Figure 1), based on the MS and NMR spectral data (Appendix A), as well as comparisons with the corresponding reports [24,25,26,27,28,29,30,31,32]. All the compounds belong to coumarins.

### 2.3. Nematicidal Activity of Compounds

The eight coumarins (compounds **1**–**8**) isolated from the chromatographed bioactive fractions of the two botanical extracts were shown to have significant nematicidal activity against PWN, with LC_50_ values ranging from 64 μM to 500 μM (Table 5). Cindimine (Compound **3**) showed the strongest anti-PWN activity, with an LC_50_ value of 64 μM in 72 h (Table 5).

### 2.4. Effect of Compounds on Egg Hatching and Reproduction

All eight coumarins inhibited the egg hatching of PWNs, with the hatching rate varying from 3.15% to 36.90% at 100 μg/mL compared with the negative control (Figure 2a). Cindimine (Compound **3**) showed the highest inhibiting effect, and another three coumarins—isoimperatorin (Compound **6**), imperatorin (Compound **7**), and bergapten (Compound **8**)—also significantly inhibited PWN egg hatching with hatching rates of less than 10%.

According to the reproduction inhibition assays, all eight nematicidal coumarins showed different inhibiting effects on PWN reproduction in contrast with the negative control at 100 μg/mL. Among them, cindimine (Compound **3**), isoimperatorin (Compound **6**), imperatorin (Compound **7**), and bergapten (Compound **8**) greatly inhibited PWN reproduction with inhibition rates of higher than 80% (Figure 2b).

### 2.5. Effect of compounds on Feeding Inhibition

Through continuously observing the feeding of PWNs in different groups, all eight nematicidal coumarins showed different inhibiting effects on the feeding capacity of PWN in contrast with the negative control (Figure 3). On the sixth day, *B. cinerea*, in Petri dishes of the negative control group, was evidently consumed by PWNs, while *B. cinerea* in the experimental groups changed little. On the eighth day, the *B. cinerea* mycelia of the negative control were almost exhausted; by comparison, there were still healthy *B. cinerea* mycelia in the experimental groups and the positive control. Especially in the groups treated by cindimine (Compound **3**), isoimperatorin (Compound **6**), imperatorin (Compound **7**), and bergapten (Compound **8**), there were much fewer *B. cinerea* mycelia consumed than those in positive control, which indicated that the four coumarins significantly inhibited the feeding capacity of PWN.

### 2.6. Effect of Compounds on AChE and Ca^2+^ ATPase

In order to explore the nematicidal compounds’ action targets against PWN, we investigated the effects of all eight coumarins on the key enzymes of PWN. The results showed that all eight nematicidal coumarins could inhibit the AChE and Ca^2+^ ATPase of PWNs at 100 μg/mL in 24 h (Figure 4).

Cindimine (Compound **3**) and imperatorin (Compound **7**) showed the strongest inhibitory abilities on AChE, with inhibition rates of 80.4% and 71.4%, respectively, which were higher than that of the positive control (Figure 4a).

According to the assays on Ca^2+^ ATPase, cindimine (Compound **3**) and imperatorin (Compound **7**) demonstrated the strongest inhibiting effects as well, with rates higher than 90% (Figure 4b). In addition, isopimpinellin (Compound **4**), isoimperatorin (Compound **6**), and bergapten (Compound **8**) also significantly inhibited the Ca^2+^ ATPase activity of PWN, with inhibition rates of more than 60%, which were much higher than that of the positive control (Figure 4b).

### 2.7. Influence on PWN Pathogenicity

To verify the influence of the nematicidal coumarins (compounds **1**–**8**) on PWN pathogenicity, pine seedling infection assays were carried out. On day 15, following inoculation with PWN, the pine seedlings in the negative control group treated with a 5% DMSO aqueous solution completely wilted (Figure 5). In contrast, the pine seedlings in the experimental groups treated with nematicidal coumarins and the positive control were much healthier (Figure 5). Specifically, the seedlings treated with cindimine (Compound **3**) showed no wilting symptoms at all, and the seedlings in the other seven experimental groups and positive control presented slight wilting symptoms on only a few leaf tips. The results indicated that the nematicidal coumarins, especially cindimine (Compound **3**), could significantly inhibit the PWN pathogenicity to the infected pine seedlings and had the potential to control PWD through trunk injection.

## 3. Materials and Methods

### 3.1. Materials and Instruments

Dried *Cnidium monnieri* (L.) Cusson fruits and *Angelica dahurica* (Fisch. *ex* Hoffm.) Benth., *et* Hook. f. *ex* Franch *et* Sav. roots were purchased from Qingdao Jianlian Pharmacy (Qingdao, China). PWNs were isolated from the branches of infected black pine trees on the campus of Qingdao University by using the Behrman funnel method [33]. The collected PWNs were washed three times with sterile water and inoculated on *Botrytis cinerea* in a potato dextrose agar (PDA) medium to be cultured at 25 °C, in darkness, for 7 d. Silica gel for chromatography (200–300 mesh) was purchased from Qingdao Ocean Chemical Co., Ltd. (Qingdao, China). PWNs were observed under a stereomicroscope (Motic BA200, Xiamen, China).

### 3.2. Isolation and Identification of Compounds

*C. monnieri* fruits (100 g) and *A. dahurica* roots (100 g) were powdered and extracted with ethanol (500 mL × 2) by the aid of intermittent ultrasonic oscillation for 24 h. The two extracted solutions were separately concentrated under a vacuum to obtain ethanol crude extracts of *C. monnieri* fruits and *A. dahurica* roots (12.5 g and 12.9 g), and then, they were partitioned between ethyl acetate and distilled water three times. The ethyl acetate-soluble and aqueous fractions of *C. monnieri* fruits and *A. dahurica* roots, respectively, were concentrated to a point of dryness.

The ethyl acetate extract of *C. monnieri* fruits (1.2 g) with nematicidal activity against PWN, according to bioassays, was fractionated by silica gel column chromatography, eluted with a stepwise gradient of hexane ethyl-acetate (5:1, 2:1, and 1:3, *v*/*v*), and 18 fractions, Fr. 1–18, were obtained. Fr. 1, Fr. 2, and Fr. 3 were confirmed to demonstrate anti-PWN activity. Fr. 1 was rechromatographed through a silica gel column, eluted with hexane-ethyl acetate (5:1, *v*/*v*), to obtain five fractions, Fr. 1.1–Fr. 1.5, among which Fr. 1.3 was found to have the strongest nematicidal activity and was recrystallized in ethyl acetate at −18 °C to yield Compound **1** (146.7 mg). Fr. 2 was purified by silica gel column chromatography eluted with hexane-ethyl acetate (3:1, *v*/*v*) to yield two active fractions: Fr. 2.1 and Fr. 2.2. Fr. 2.1 was rechromatographed through a silica gel column eluted with hexane-ethyl acetate (3:1, *v*/*v*) to yield Compound **1** (10 mg), Compound **2** (31 mg), and Compound **3** (21.7 mg). Fr. 2.2 was also further separated through a silica gel column eluted with hexane-ethyl acetate (3:1, *v*/*v*) to yield Compound **4** (13.7 mg). In addition, Fr. 3, as a nematicidal component, was recrystallized in ethyl acetate at −18 °C to yield Compound **5** (12.02 mg).

The nematicidal ethyl acetate extract of *A. dahurica* roots (2.4 g) was subjected to silica gel column chromatography eluted with a stepwise gradient of hexane-ethyl acetate (2:1, 1:1, 1:5, *v*/*v*, and 100% ethyl acetate,) to acquire 16 fractions: Fr. 1–16. The three active fractions, Fr. 5–Fr. 7, were further purified. Compound **6** (83.8 mg) and Compound **7** (233.4 mg) were separately obtained from Fr. 5 and Fr. 6 using silica gel column chromatography eluted with hexane-ethyl acetate (3:1, *v*/*v*). Fr. 7 was chromatographed through a silica gel column eluted with hexane-ethyl acetate (3:1, *v*/*v*) to obtain Compound **1** (2.6 mg), Compound **2** (3.7 mg), Compound **4** (5.2 mg), and Compound **8** (8.4 mg).

The chemical structures of compounds **1**–**8** were identified by ^1^H NMR at 600 MHz and ^13^C NMR at 150 MHz in chloroform-d (CDCl_3_) or dimethylsulfoxide-d_6_ (DMSO-d_6_), with tetramethylsilane (TMS) as the internal standard, using a nuclear magnetic resonance spectrometer (NMR, JNM-ECZ600R, JEOL, Tokyo, Japan), and electron impact mass spectrometry (EIMS) was determined by gas chromatography-mass spectrometry (GC–MS, 7890A-5975C, Agilent, Santa Clara, CA, USA).

### 3.3. Nematicidal Assay

The extracts and chromatographed fractions of *C. monnieri* fruits and *A. dahurica* roots were separately dissolved in dimethylsulphoxide (DMSO) to prepare test solutions. Compounds **1**–**8** were individually dissolved in DMSO and, then, successively diluted to a series of concentrations (1–4 mg/mL) representing the test solutions. Test solutions (5 μL each) were introduced into 96-well plates containing around 100 pine wood nematodes in 95 μL 0.5% Triton X-100 aqueous solution. A 5% DMSO aqueous solution containing 0.5% Triton X-100 was used as the negative control, and aloperine, a plant-derived nematicide which has gone through field tests for PWN control [34,35], was used as the positive control. Each treatment was repeated three times. The numbers of dead and live nematodes were recorded, after incubation at 25 °C in darkness, for 24 h, 48 h, and 72 h. The nematodes were observed under a stereomicroscope and considered dead if their bodies were rigid and did not respond to physical stimuli from a fine needle [36,37,38]. The corrected mortality of nematodes was calculated according to the following formula [38]:Corrected mortality% = (Mortality% in treatment − Mortality% in negative control)/(1 − Mortality% in negative control)
(1)

### 3.4. Egg Hatching Inhibition Assay

PWN eggs were obtained according to the method reported by Liu et al. [39]. To evaluate the influence of the compounds **1**–**8** on the egg hatching of PWN, about 100 PWN eggs were transferred to each test well of a 48-well plate, containing different nematicidal compounds dissolved in 5% DMSO aqueous solution, as well as containing 0.5% Triton X-100 at a final concentration of 100 μg/mL. A 5% DMSO aqueous solution, containing 0.5% Triton X-100 and aloperine, was used as the negative and positive control, respectively. Each treatment was repeated three times and operated at 25 °C for 24 h. The hatching rate was calculated as follows:Hatching rate% = Larva/(Egg + Larva) × 100%
(2)

### 3.5. Reproduction Inhibition Assay

Approximately 150 nematodes were separately soaked in 100 μL solutions of nematicidal compounds **1**–**8** (100 μg/mL) and incubated at 25 °C for 24 h. Nematodes soaked in 5% DMSO aqueous solution, containing 0.5% Triton X-100 and aloperine solution (100 μg/mL), were separately used as the negative and positive controls. After that, the nematodes in each group were individually transferred to a Petri dish fully covered with *B. cinerea* on a PDA medium and cultured at 25 °C, in darkness, for 8 d. Then, the nematodes were isolated and washed four times with sterile distilled water. The numbers of nematodes were counted under a stereomicroscope [39]. Each test was repeated three times. The reproduction inhibition rate was calculated according to the following formula:Reproduction inhibition rate% = (1 − Final PWN population in treated group/Final PWN population in negative control) × 100%
(3)

### 3.6. Feeding Inhibition Assay

Around 150 nematodes were introduced to the different nematicidal compound solutions (100 μg/mL) in the 96-well plate and cultured at 25 °C, in the dark, for 24 h. Nematodes soaked in a 5% DMSO aqueous solution, containing 0.5% Triton X-100 and the aloperine solution (100 μg/mL), were taken as the negative and positive controls, respectively. Each treatment was repeated three times. Each group of treated nematodes was washed three times using sterile water and transferred to a Petri dish fully covered with *B. cinerea*, cultured on PDA plates, and incubated at 25 °C in the dark. The feeding of PWNs was continuously observed and recorded.

### 3.7. Effects of the Compounds on the Enzymes of PWNs

#### 3.7.1. Preparation of Enzyme Solution

Around 2000 healthy PWNs were washed three times with sterile water and homogenized in 1 mL iced physiological saline. The homogenate was centrifuged at 4 °C, 12,000 rpm for 10 min, and the supernatant was collected as the enzyme solution used for assays. The protein concentration of PWN enzyme solution was determined by Bradford’s method, with bovine serum albumin as the standard [40].

A 5 μL DMSO solution of each nematicidal compound (2 mg/mL) was separately mixed with 95 μL of the nematode enzyme solution to obtain the treated enzyme solutions. The Enzyme solutions treated with 5% DMSO and aloperine, at the same concentration as the isolated nematicidal compounds, were used as negative and positive controls, respectively. The above treated enzyme solutions were incubated at 37 °C for 15 min, and each treatment was repeated three times.

#### 3.7.2. Effect on AChE

AChE activity was measured by using the improved method reported by Ellman et al. [41,42,43]. A total 200 μL of enzyme solution and 100 μL acetylthiocholine iodide (ATCH, 2 mM) in 0.1 M phosphate buffer (PBS, pH 7.2) were mixed and incubated at 30 °C for 30 min. A total 100 μL of 5,5′-dithiobis (2-nitrobenzoic acid) developer (10 mM) was then added, and the reaction solution was continually incubated for 6 min. After that, 500 μL of 4% SDS was introduced to stop the color reaction [44]. Finally, 1 mL PBS (0.1 M, pH 7.2) was added to the reaction system, and OD_412_ was recorded on a UV–Vis spectrophotometer (GENESY-50, Thermo Fisher Scientific, Waltham, MA, USA). Among AchE activity (U), one unit was defined as hydrolyzing 1 nmol of acetylcholine per mg of protein in one minute under assay conditions. The inhibition rate of AchE activity was calculated as follows [45]:Inhibition rate% = (U_negative control_ − U_test sample_)/U_negative control_ × 100%(4)

#### 3.7.3. Effect on Ca^2+^ ATPase

The activity of Ca^2+^ ATPase, in each enzyme solution, was tested by using a Ca^2+^ ATPase assay kit (Nanjing Jiancheng Bioengineering Institute, Nanjing, China). Among Ca^2+^ ATPase activity, one unit was defined as 1 μmol inorganic phosphorus produced by ATP decomposition and catalyzed by Ca^2+^ ATPase per mg protein in one hour. The inhibition rate of Ca^2+^ ATPase activity was calculated as follows:Inhibition rate% = (U_negative control_ − U_test sample_)/U_negative control_ × 100%(5)

### 3.8. Influence of the Compounds on Pathogenicity to Pine Seedlings

The 30 day-old sterile black pine seedlings were prepared according to the method reported by Sun et al. [11]. A total of 10 μL nematode suspension (15 PWNs/μL) was dripped into pine seedlings using a syringe [46]. On the second day following the inoculation of PWN, 10 μL of each nematicidal compound solution (100 µg/mL) was introduced to each seedling of the corresponding experimental group using a syringe. In contrast, the same amount of solvent used to dissolve the nematicidal compounds, namely a 5% DMSO aqueous solution, was injected into each seedling of the negative control, and the same volume of the aloperine solution (100 µg/mL) was injected into each seedling of the positive control. Each treatment was repeated three times.

### 3.9. Statistical Analysis

All of the data processing was based on SPSS version 25.0 software. The median lethal concentration (LC_50_) was obtained by probit analysis.

## 4. Discussion

The search for eco-friendly nematicides to enrich alternatives for PWD management has recently become an urgent research topic. In recent years, we have found several plant-derived coumarins with potential for PWN control [47,48], which prompted us to explore the nematicidal activity of coumarins from plant secondary metabolites. In this research, we confirmed that the ethyl acetate extracts of two plant materials rich in coumarins, *C. monnieri* fruits and *A. dahurica* roots, displayed significant anti-PWN activity for the first time, and we identified their main nematicidal compounds, through bioactivity-guided fractionation, to be eight coumarins (compounds **1**–**8**)**,** among which cindimine (Compound **3**), isopimpinellin (Compound **4**), and marmesin (Compound **5**) were first reported to have nematicidal activity. Cindimine (Compound **3**), especially, showed the highest anti-PWN activity and the strongest potential for inhibiting PWN pathogenicity.

The inhibitory effect of the eight identified coumarins**,** especially compounds **3** and **6**–**8**, on the egg hatching, reproduction, and feeding of PWN further demonstrated their anti-PWN capacity. Notably, egg hatchability is considered one of the important parameters affecting the PWN population, and any chemical substance with high embryonic lethality could be a potential nematicidal agent [49]. This research provided more evidence for the anti-PWN activity of coumarins, as well as helpful clues regarding the structure–activity relationship. The core moiety of coumarins, 2*H*-1-Benzopyran-2-one, is a critically effective structure since the main active compounds of the two botanical extracts all belong to coumarins. Furthermore, the ester group and α,β-unsaturated carbonyl in the side chain of cindimine (Compound **3**), the most potent nematicidal coumarin discovered in the research, could contribute to nematotoxicity, which coincides with our previous report on the excellent nematicidal coumarin columbianadin [48].

In exploring the nematicidal mechanism of the identified coumarins against PWN, we found that all eight coumarins, especially cindimine (Compound **3**) and imperatorin (Compound **7**), displayed significant inhibitory effects on the AChE and Ca^2+^ ATPase of PWNs. AchE, as an important enzyme in most vertebrates, insects, and nematodes with a cholinergic nervous system, catalyzes the hydrolysis of the neurotransmitter acetylcholine (ACh) and terminates nerve impulses [50]. Inhibition of AChE can lead to the accumulation of ACh in the synaptic cleft and overexcitation of cholinergic neurons, resulting in insect mortality, and many pesticides, such as organophosphates and carbamates, are known to affect the activity of AChE [51,52]. Ca^2+^ ATPases, as ubiquitously distributed membrane enzymes, maintain calcium homeostasis inside eucaryotic cells and generally counteract the influx of free Ca^2+^ ions through calcium channels to exert an essential role in controlling enzymatic reactions and a broad spectrum of intracellular signaling processes [53,54]. Since AChE and Ca^2+^ ATPases both exert important physiological functions in organisms, the two key enzymes of PWN could be the critical action targets of the nematicidal coumarins against PWN.

## 5. Conclusions

In this research, the two ethanol extracts of *C. monnieri* fruits and *A. dahurica* roots were validated to have significant nematicidal activity against PWN, and eight active coumarins, osthol (Compound **1**), xanthotoxin (Compound **2**), cindimine (Compound **3**), isopimpinellin (Compound **4**), marmesin (Compound **5**), isoimperatorin (Compound **6**), imperatorin (Compound **7**), and bergapten (Compound **8**), were isolated from the two plant extracts. Among them, cindimine (Compound **3**) showed the highest nematicidal activity against PWN. Compounds **3** and **6**–**8**, especially compounds **3** and **7**, exhibited stronger inhibitory effects on the egg hatching, feeding ability, and reproduction of PWN, as well as the two critical enzymes of PWN, AChE and Ca^2+^ ATPase. Furthermore, we found that the eight nematicidal coumarins, especially cindimine (Compound **3**), could markedly weaken PWN pathogenicity on infected pine seedlings, which indicated that the coumarins had the potential to be eco-friendly nematicides or lead compounds of synthesized derivatives with enhanced activity for sustainable PWD control through trunk injection. Cindimine (Compound **3**) had the strongest nematicidal activity and, therefore, deserves more interest.

## Figures and Tables

**Figure 1 molecules-28-04109-f001:**
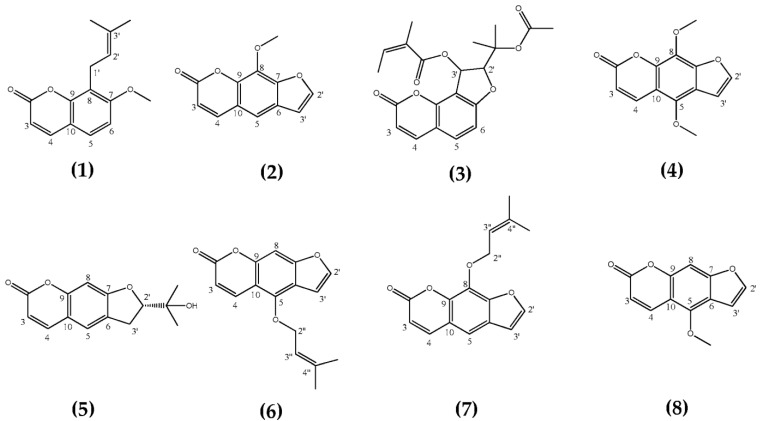
Chemical structures of the nematicidal coumarins (compounds **1**–**8**).

**Figure 2 molecules-28-04109-f002:**
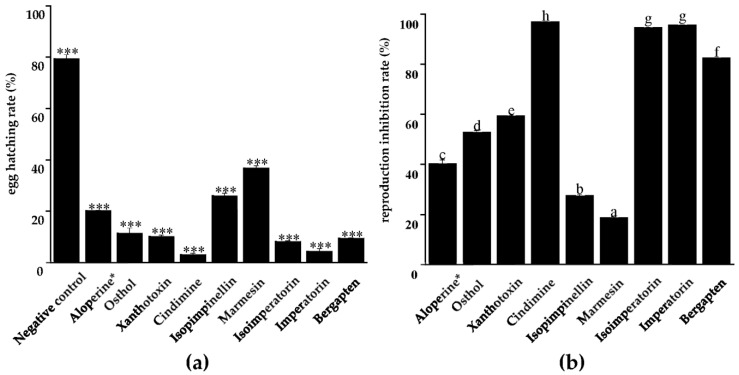
The egg hatching rate (**a**) and reproduction inhibition rate (**b**) of the eight nematicidal coumarins. The asterisks denote significant differences at *p* < 0.001 compared to negative control, according to Student’s *t*-test. Different letters mean significant differences at *p* < 0.05 using the LSD test.

**Figure 3 molecules-28-04109-f003:**
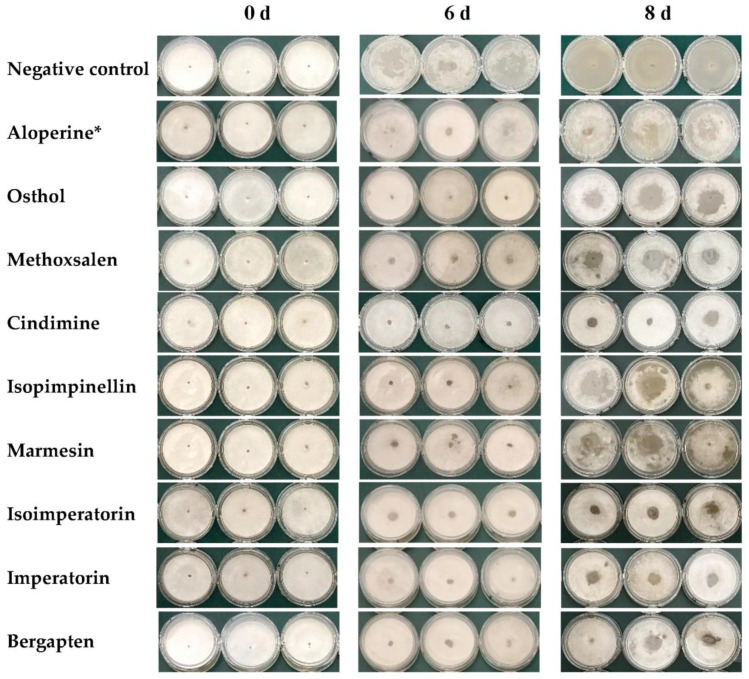
The effect of nematicidal coumarins on the feeding capacity of PWN. * represents positive control.

**Figure 4 molecules-28-04109-f004:**
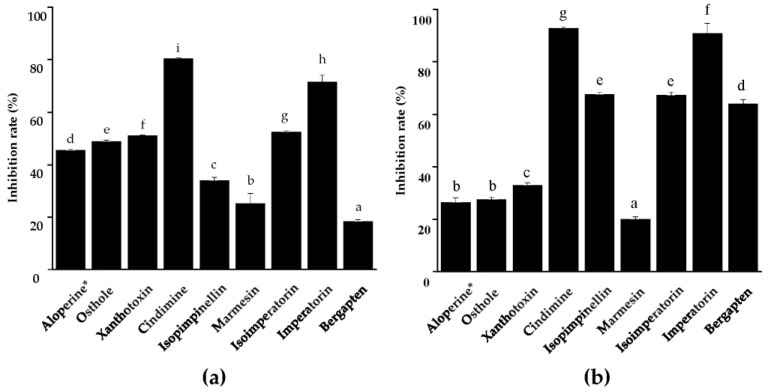
Inhibition rates of nematicidal coumarins on AChE (**a**) and Ca^2+^ ATPase (**b**) of PWNs. Means with different letters were significantly different at *p* < 0.05 based on the LSD test. * represents positive control.

**Figure 5 molecules-28-04109-f005:**
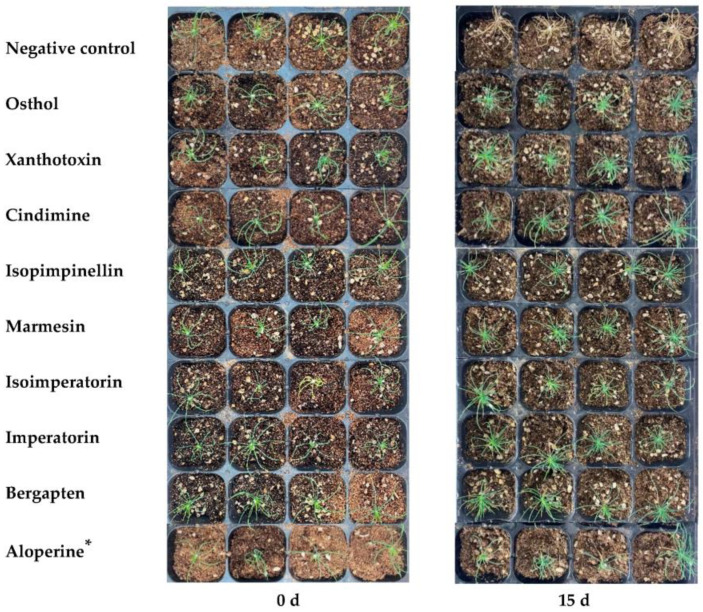
Effect of the coumarins on Japanese black pine seedlings infected by PWN. * represents positive control.

**Table 1 molecules-28-04109-t001:** Nematicidal activities of the *C. monnieri* fruit extracts.

Samples (1 mg/mL)	Corrected Mortality (%, Mean ± SD)		
	24 h	48 h	72 h
Ethanol extract	43.14 ± 0.20	62.08 ± 0.41	77.93 ± 0.82 b
Ethyl acetate extract	78.30 ± 2.49	82.33 ± 2.49	96.00 ± 0.81 c
Aqueous fraction	5.23 ± 0.33	12.3 ± 0.81	14.01 ± 2.49 a

Means in the column followed by the same letter did not differ significantly (*p* = 0.05) in 72 h, according to the LSD test.

**Table 2 molecules-28-04109-t002:** Nematicidal activities of the *A. dahurica* root extracts.

Samples (1 mg/mL)	Corrected Mortality (%, Mean ± SD)		
	24 h	48 h	72 h
Ethanol extract	32.28 ± 0.99	65.60 ± 0.50	84.64 ± 0.55 b
Ethyl acetate extract	29.53 ± 1.68	66.81 ± 1.31	86.15 ± 0.92 c
Aqueous fraction	7.38 ± 1.22	16.2 ± 0.83	20.27 ± 0.83 a

The mean corrected mortalities of nematodes followed by the same letter did not differ significantly (*p* = 0.05) in 72, h according to the LSD test.

**Table 3 molecules-28-04109-t003:** Nematicidal activities of the chromatographed fractions from *C. monnieri* fruit extract.

Sample (500 µg/mL)	Corrected Mortality (%, Mean ± SD)		
	24 h	48 h	72 h
Fr.1	71.14 ± 1.24	85.86 ± 0.82	93.92 ± 0.85 n
Fr.2	76.51 ± 1.29	83.84 ± 0.82	91.22 ± 1.23 m
Fr.3	53.69 ± 0.85	61.28 ± 0.48	69.60 ± 0.71 l
Fr.4	12.08 ± 0.06	24.58 ± 0.48	30.41 ± 0.84 h
Fr.5	1.68 ± 0.47	9.43 ± 0.48	13.51 ± 0.45 de
Fr.6	0.67 ± 0.47	2.36 ± 0.48	3.04 ± 0.01 a
Fr.7	13.76 ± 0.45	27.61 ± 0.48	40.20 ± 0.87 i
Fr.8	19.80 ± 0.44	23.91 ± 0.48	41.55 ± 0.66 j
Fr.9	5.70 ± 0.46	14.81 ± 0.95	29.39 ± 0.14 h
Fr.10	3.68 ± 0.92	6.10 ± 0.82	49.33 ± 0.63 k
Fr.11	1.67 ± 0.94	3.37 ± 0.48	5.40 ± 0.46 b
Fr.12	2.01 ± 0.01	6.06 ± 0.82	29.73 ± 0.42 h
Fr.13	4.70 ± 0.46	9.43 ± 0.48	14.53 ± 0.45 g
Fr.14	0.67 ± 0.47	5.38 ± 0.95	12.50 ± 0.90 d
Fr.15	1.34 ± 0.48	11.45 ± 0.48	13.50 ± 0.89 ef
Fr.16	4.70 ± 0.46	6.73 ± 0.95	14.18 ± 1.60 fg
Fr.17	0.67 ± 0.47	1.34 ± 0.48	2.70 ± 0.95 a
Fr.18	1.68 ± 0.47	2.69 ± 0.48	7.43 ± 0.46 c

Means of the corrected mortality at 72 h followed by the same letter did not differ significantly (*p* = 0.05), according to the LSD test.

**Table 4 molecules-28-04109-t004:** Nematicidal activities of the chromatographed fractions from *A. dahurica* roots.

Sample (500 µg/mL)	Corrected Mortality (%, Mean ± SD)		
	24 h	48 h	72 h
Fr.1	6.67 ± 0.58	39.24 ± 1.30	59.65 ± 1.31 i
Fr.2	3.34 ± 0.58	9.72 ± 1.30	21.75 ± 0.99 c
Fr.3	2.67 ± 1.15	17.01 ± 0.49	28.42 ± 1.72 e
Fr.4	2.33 ± 0.58	17.36 ± 0.49	26.32 ± 2.27 d
Fr.5	2.33 ± 0.58	32.64 ± 0.49	70.87 ± 0.50 j
Fr.6	2.33 ± 0.58	17.36 ± 0.98	74.74 ± 1.49 k
Fr.7	9.67 ± 0.58	44.79 ± 0.85	79.65 ± 3.47 l
Fr.8	9.67 ± 0.58	39.58 ± 1.70	49.82 ± 2.16 g
Fr.9	10.33 ± 0.65	45.14 ± 0.49	53.68 ± 3.10 h
Fr.10	9.33 ± 0.58	18.75 ± 1.47	28.77 ± 0.49 e
Fr.11	2.33 ± 0.58	13.19 ± 0.49	17.89 ± 1.72 b
Fr.12	4.33 ± 0.58	13.89 ± 0.49	20.00 ± 0.86 c
Fr.13	6.33 ± 0.58	18.75 ± 0.85	31.92 ± 0.50 f
Fr.14	6.33 ± 0.58	18.40 ± 1.30	29.82 ± 2.48 e
Fr.15	3.67 ± 0.58	12.15 ± 0.98	14.39 ± 1.31 a
Fr.16	2.33 ± 0.58	33.68 ± 0.49	48.07 ± 0.99 g

Means of the corrected mortality at 72 h followed by the same letter did not differ significantly (*p* = 0.05), according to the LSD test.

**Table 5 molecules-28-04109-t005:** LC_50_ values of the compounds isolated from *C. monnieri* fruits and *A. dahurica* roots against PWN at 72 h.

	LC_50_ (μg/mL)	LC_50_ (μM)	95% Confidence Intervals (μg/mL)	Χ^2^
Aloperine *	89.90	387	78.01–104.99	4.34
Osthol (Compound **1**)	64.93	266	56.10–74.08	4.95
Xanthotoxin (Compound **2**)	54.68	253	46.55–63.08	1.76
Cindimine (Compound **3**)	24.73	64	11.15–36.01	0.98
Isopimpinellin (Compound **4**)	92.16	375	80.44–111.24	2.85
Marmesin (Compound **5**)	122.96	500	111.03–136.69	0.65
Isoimperatorin (Compound **6**)	43.08	160	31.44–53.91	0.64
Imperatorin (Compound **7**)	35.72	132	26.62–44.13	3.07
Bergapten (Compound **8**)	52.07	241	45.59–62.16	2.90

* positive control in the experiment.

## Data Availability

The data presented in this study are available in article and Appendix A.

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
