# Peer review of "Nematicidal Coumarins from Cnidium monnieri Fruits and Angelica dahurica Roots and Their Physiological Effect on Pine Wood Nematode (Bursaphelenchus xylophilus)"

_molecules, 2023, doi:10.3390/molecules28104109_

Round 1

Reviewer 1 Report (New Reviewer)

This manuscript reports the identification of nematicidal coumarins from Cnidium monnieri fruits and Angelica dahunica roots, with their physiological effect on pine wood nematode.  The content has originality and novelty, and this manuscript has interest to the readers.  However, the following issues would be needed before accept. 

1) p.3, line 110, "100 MHz":   If the spectrometer used is JNM-ECZ600R, 13C NMR would be "150 MHz".  "100 MHz" is correct? 

2) p.7, line 237-p.8, line 296:   Eight compounds 1-8 are all known and identified by comparing with the literature data.  Thus, it will be better to remove the spectral data.  Indication of the references ([38]-[46]) would be enough. 

3) LC50 values:   The potency (LC50 value) of the isolated compounds should be expressed with mM unit, not with mg/mL.  If it is needed to compare the activity of the extracts (expressed with mg/mL unit), both values should be indicated. 

4) The following spell should be corrected. 

p.3, line 112:  "jnm-ecz600r" to "JNM-ECZ600R"

p.3, line 112:  "joel" to "JEOL"

p.3, line 114:  "7890a-5975c" to "7890A-5975C

p.12, line 386:  "2H-1-benzopyran-2-one" to "2H-1-Benzopyran-2-one"  ("H" of "2H" should be italicized.  "b" of "benzopyran should be capital.) 

p.12, line388: "a, b-" to "a,b-"  (Remove the space between "a," and "b-".) 

p.13, line 398:  "AchE" to "AChE"

The following spell should be corrected. 

p.3, line 112:  "jnm-ecz600r" to "JNM-ECZ600R"

p.3, line 112:  "joel" to "JEOL"

p.3, line 114:  "7890a-5975c" to "7890A-5975C

p.12, line 386:  "2H-1-benzopyran-2-one" to "2H-1-Benzopyran-2-one"  ("H" of "2H" should be italicized.  "b" of "benzopyran should be capital.) 

p.12, line388: "ab-" to "a,b-"  (Remove the space between "a," and "b-".) 

p.13, line 398:  "AchE" to "AChE"

Author Response

Reviewer 2 Report (New Reviewer)

The authors describe the isolation of eight nematicidal coumarins from the ethyl acetate extracts of C. monnieri fruits and A. dahurica roots through bioassay-guided fractionations. The isolated compounds were identified by NMR and mass spectroscopy. The authors claimed that the isolated compounds showed nematicidal activity against Pine wood nematode (PWN) and demonstrated in bioassays to relieve the wilt symptoms of black pine seedlings infected by PWN. The experimental part is informative and clear. The chromatographic separation of the isolated compounds was clearly explained and discussed. The compounds were identified by NMR and mass spectroscopy. The authors used various biological tests to identify the PWN nematicidal activity of the isolated compounds including enzyme binding assay and bioassay and were all well designed and supportive. The discussion part was well written, and the conclusion drawn was reliable. The manuscript is over all organized and uses appropriate language.

This work merits publication after minor corrections.

1. The author did not give any information about the relative amount of isolates from each plant used and whether they both contains the 8 isolated compounds and in what relative yield. This would affect the recommendation on using those plants as nemitocides.

2. The difference in the mortality from the crude ethyl acetate extract is obvious in both plants (table 1 & table 3). The discussion should include a clarification for that.

3. The author is describing the NMR and mass spectroscopy data in the discussion. This part should be moved to the experimental or appendix.

4. The compounds were previously reported and 1HNMR would be sufficient to identify the compounds with comparison to reference spectra.

5. The purity data (HRMS, elemental, HPLC) for the isolated compounds are lacking and this an essential part since the compounds were biologically tested and claimed to be active.

6. The author did not give any information about the previously reported activities of the isolated compounds and whether they were previously proven to possess any antiparasitic or antimicrobial activities. 

Author Response

This manuscript is a resubmission of an earlier submission. The following is a list of the peer review reports and author responses from that submission.

Round 1

Reviewer 1 Report

Its can be publish with minor revision

please correct plant name 

Author Response

Q: please correct plant name

A: Thanks a lot for the reviewer’s kind suggestions and careful revisions. We have revised all the plant names except Fructus cnidii because we isolated some reported active compounds exactly from Fructus cnidii , the seed of ‘ Cindium mommieri (L.) Cusson’ that the reviewer suggested. In addition, we also corrected other errors the reviewer pointed out.

Reviewer 2 Report

The research offers some plant-derived nematicidal coumarins against PWN to control PWD, a devastating pine disease causing severe worldwide damage to susceptible pine species and forest ecosystems, and studied their physiological effects on PWN as well. It is meaningful and interesting for the researchers in this field. The work is all-round and welldone. However, there are some minor points to revise or respond:

1. Line 5, Behrman →Baermann

2. Line 69 with→under

3. Line 102, what is the aforementioned method? There are several aforementioned elution schemes, so here the detailed elution method is needed.

4. Please Check the fomula (3). I think it’s unreasonable.

5. Line199&202, morality→mortality

6. L359 acitivity→activity

7. L386-387 action target against PWN of the nematicidal coumarins→action target of the nematicidal coumarins against PWN

Author Response

Q1&2: Line 5, Behrman →Baermann;Line 69 with→under

A: Thanks. We have revised the two words.

Q3: Line 102, what is the aforementioned method? There are several aforementioned elution schemes, so here the detailed elution method is needed.

A: Thanks, yes, we added the detailed method in the corresponding section.

Q4: Please Check the fomula (3). I think it’s unreasonable.

A:Yes, there is an error in this fomula, and we have revised it. Thanks.

Q5&6: Line199&202, morality→mortality; L359 acitivity→activity

A: Thanks. We have revised these words.

Q7: L386-387 action target against PWN of the nematicidal coumarins→action target of the nematicidal coumarins against PWN

A: Thanks. We have revised the phrase.

Reviewer 3 Report

The authors isolated and identified eight nematicidal coumarins from two plant extracts, and studied their biological activity and mode of action against pine wood nematode. The experiments are well designed and the work is described clearly. This manuscript will be very helpful for researchers to investigate new eco-friendly drugs against PWN. I still have some minor comments:

1. Line 5-- Behrman funnel should be ‘Baermann funnel’?

2. Line 99-- what are the three active fractions? They need to be pointed out.

3. Line 115,116-- Test solutions (5 μl) were...--->Each test solution (5 μl) was

4. Please explain why difference analysis among the data in the columns of 24 h and 48 h (table 1-4) aren’t carried out.

5. L363-364 Delete ‘numbers’ in ‘number of PWN population numbers’

6. For No.6 in Reference list, page numbers should be given.

Author Response

For the reviewer’s suggestion on English language editing, we had our mancuscript further checked by a native speaker from USA and improved some sentences, Thanks.

For the other questions,

Q1: Line 5-- Behrman funnel should be ‘Baermann funnel’?

A: Yes, Thanks. We have revised ‘Behrman’ to ‘Baermann’.

Q2: Line 99-- what are the three active fractions? They need to be pointed out.

A: Thanks. We added the names of the three active fractions.

Q3: Line 115,116-- Test solutions (5 μl) were...--->Each test solution (5 μl) was

A: Thanks. We have revised this sentence.

Q4: Please explain why difference analysis among the data in the columns of 24 h and 48 h (table 1-4) aren’t carried out.

A: Thanks. We only compared the data in the column of 72 h (table 1-4) because we screened the active extracts or fractions according to their nematicidal activity at 72 h.

Q5: L363-364 Delete ‘numbers’ in ‘number of PWN population numbers’

A: Thanks. We have revised this phrase.

Q6: For No.6 in Reference list, page numbers should be given.

A: Thanks. We added the page numbers of the reference.